# Genome-Wide Identification of *GATA* Family Genes in Potato and Characterization of *StGATA12* in Response to Salinity and Osmotic Stress

**DOI:** 10.3390/ijms252212423

**Published:** 2024-11-19

**Authors:** Xi Zhu, Huimin Duan, Ning Zhang, Yasir Majeed, Hui Jin, Wei Li, Zhuo Chen, Shu Chen, Jinghua Tang, Yu Zhang, Huaijun Si

**Affiliations:** 1Key Laboratory of Tropical Fruit Biology, Ministry of Agriculture and Rural Affairs, South Subtropical Crops Research Institute, Chinese Academy of Tropical Agricultural Sciences, Zhanjiang 524091, China; zhuxi@catas.cn (X.Z.); dhm_80@163.com (H.D.); jinh@catas.cn (H.J.); liwei0376845169@126.com (W.L.); chenzhuo@catas.cn (Z.C.); chenshu666@catas.cn (S.C.); cotton111@163.com (J.T.); 2Key Laboratory of Hainan Province for Postharvest Physiology and Technology of Tropical Horticultural Products, South Subtropical Crops Research Institute, Chinese Academy of Tropical Agricultural Sciences, Zhanjiang 524091, China; 3National Key Laboratory for Tropical Crop Breeding, Sanya Research Institute, Chinese Academy of Tropical Agricultural Sciences, Sanya 572025, China; 4State Key Laboratory of Aridland Crop Science, Gansu Agricultural University, Lanzhou 730070, China; ningzh@gsau.edu.cn (N.Z.); yasirmajeed5453@gmail.com (Y.M.); 5College of Life Science and Technology, Gansu Agricultural University, Lanzhou 730070, China; 6College of Agronomy, Gansu Agricultural University, Lanzhou 730070, China

**Keywords:** GATA gene family, potato, salinity stress, osmotic stress

## Abstract

GATA factors are evolutionarily conserved transcription regulators that are implicated in the regulation of physiological changes under abiotic stress. Unfortunately, there are few studies investigating the potential role of *GATA* genes in potato plants responding to salt and osmotic stresses. The physicochemical properties, chromosomal distribution, gene duplication, evolutionary relationships and classification, conserved motifs, gene structure, interspecific collinearity relationship, and cis-regulatory elements were analyzed. Potato plants were treated with NaCl and PEG to induce salinity and osmotic stress responses. qRT-PCR was carried out to characterize the expression pattern of StGATA family genes in potato plants subjected to salinity and osmotic stress. *StGATA12* loss-of-function and gain-of-function plants were established. Morphological phenotypes and growth were indicated. Photosynthetic gas exchange was suggested by the net photosynthetic rate, transpiration rate, and stomatal conductance. Physiological indicators and the corresponding genes were indicated by enzyme activity and mRNA expression of genes encoding CAT, SOD, POD, and P5CS, and contents of H_2_O_2_, MDA, and proline. The expression patterns of StGATA family genes were altered in response to salinity and osmotic stress. StGATA12 protein is located in the nucleus. *StGATA12* is involved in the regulation of potato plant growth in response to salinity and osmotic stress. Overexpression of *StGATA12* promoted photosynthesis, transpiration, and stomatal conductance under salinity and osmotic stress. *StGATA12* overexpression induced biochemical responses of potato plants to salinity and osmotic stress by regulating the levels of H_2_O_2_, MDA, and proline and the activity of CAT, SOD, and POD. *StGATA12* overexpression induced the up-regulation of *StCAT*, *StSOD*, *StPOD*, and *StP5CS* against salinity and osmotic stress. *StGATA12* could reinforce the ability of potato plants to resist salinity and osmosis-induced damages, which may provide an effective strategy to engineer potato plants for better adaptability to adverse salinity and osmotic conditions.

## 1. Introduction

Soil salinization is defined as the upward movement of salinity from the lower soil layers or groundwater to the surface via capillary action [1]. This phenomenon, resulting from a combination of human and natural factors, threatens nearly 20% of the world’s irrigated agriculture [2]. Salinity persists in various cultivated areas due to insufficient efforts to address the issue. Additionally, arid regions cover 41% of the world’s land area and are inhabited by 38% of the global population [3]. Soil salinization is particularly severe in drylands, where 30% of the water used for agricultural irrigation is saline, leading to reduced crop yields, land degradation, and loss of biodiversity [4]. According to global saline soil distribution data released by the Food and Agriculture Organization (FAO) in October 2021, over 833 million hm^2^ of the world’s soils have been affected by salinization, with more than 10% of agricultural land classified as saline. These figures are on the rise, and soil salinization has become a major threat to crop production and productivity.

Potato (*Solanum tuberosum* L.) is regarded by the FAO as a key crop for ensuring food security, which is severely affected by salinity [2]. Growth of potato plants is inhibited under salt stress due to ion toxicity caused by osmotic stress [5]. Salt-induced osmotic stress often elicits complex and multifaceted physiological responses [6]. NaCl, as a major abiotic stressor, inhibits plant growth by altering transpiration rate, stomatal conductance, net photosynthetic rate, and other thermal parameters [7]. Salt stress results in stomatal closure, which reduces the ratio of CO_2_ to O_2_ in leaves and inhibits CO_2_ fixation [8]. Under osmotic stress, the production of reactive oxygen species such as superoxide radicals, hydrogen peroxide, and hydroxyl radicals increases due to enhanced electron leakage to oxygen [9]. These reactive oxygen species can damage membranes and other essential macromolecules, including photosynthetic pigments [10]. In response to salinity stress, various physiological components, such as antioxidant enzymes, are produced to mitigate this damage [11,12].

To prevent further salinization, a drainage system is established to enhance soil permeability and functionality by leaching out salts, thereby promoting better root development [13]. Additionally, another approach to mitigate the negative effects of soil salinization on agriculture is to identify crops and crop varieties that are salt tolerant. Conventional breeding is a well-established method for genetically improving crops by combining desirable traits from different varieties or species. However, developing salt-tolerant varieties through conventional breeding is challenging due to the difficulty in identifying clearly defined physiological traits associated with salt tolerance and the involvement of multiple genes. The integration of molecular tools into breeding programs is accelerating the development of salt-tolerant crop varieties with high yields and desirable agronomic traits. Genetic engineering has been successfully employed to produce salt-tolerant plants, either by overexpressing or introducing selected genes into elite cultivars. Numerous transgenic studies have focused on improving salt tolerance in potato using genes from various sources [14,15,16].

GATA factors are evolutionarily conserved transcription regulators that are found in animals, fungi, and plants, which recognize the DNA sequence W-G-A-T-A-R through a single type IV zinc finger. Genome-wide identification and characterization of GATA family genes have been reported in Arabidopsis [17], Rubiaceae [18], tomato, potato [19], rapeseed [20], and cucumber [21]. In recent years, knowledge about the biological role and regulation of plant GATAs has substantially improved. Individual family members have been implicated in the regulation of chlorophyll biosynthesis, glucose sensitivity [17], flower development [22], chloroplast division [23], photosynthesis [24], light and circadian response [25], et al. GATA factors also respond to diverse abiotic stresses in plants, which have been investigated under salinity, drought, exogenous ABA, high temperature, and low nitrogen conditions [21,26,27]. Unfortunately, there are few studies investigating the potential role of *GATAs* in potato plants responding to salinity and osmotic stresses.

In this study, we aimed to investigate the expression patterns of *StGATA* genes in response to salt and osmotic stress. We then selected the responsive genes for further validation of their roles following NaCl and PEG6000 treatment. Our findings may provide insights into the molecular mechanisms underlying plant tolerance to salt and osmotically induced oxidative stress.

## 2. Results

In this study, we identified 57 *StGATA* genes in potato plants and classified them into four subfamily groups. The *StGATA* genes exhibited organ-specific expression patterns and showed altered expression in response to salinity and osmotic stresses. Notably, *StGATA12* was found to modulate photosynthesis, transpiration, and stomatal conductance under these stress conditions. Furthermore, overexpression of *StGATA12* enhanced the biochemical responses of potato plants to salinity and osmotic stress by regulating levels of H_2_O_2_, MDA, and proline, as well as the activities of CAT, SOD, and POD. It also induced genetic responses, including the expression of *StCAT*, *StSOD*, *StPOD*, and *StP5CS*, thereby enhancing the plant’s resilience to salinity and osmotic stress. The study design is presented in Figure 1.

### 2.1. Identification of GATA Gene Family in Potato

Through searching against Arabidopsis thaliana, we identified 57 GATA family members in potato. The protein features and subcellular location of GATA proteins in potato are depicted in Appendix A. We found the distribution of 57 *StGATAs* on chromosomes 1–10. The fifty-seven StGATA protein members exhibit different protein properties. The protein length of StGATA members ranges between 81 and 543 aa, and protein weights differ between 8782.43 and 60,626.92 Da. The isoelectric points range from 3.38 to 10.34. The GRAVY values of StGATA proteins are negative, ranging from −1.056 to −0.221. All *GATA* gene-coding proteins were located in the nucleus. Particularly, StGATA54 was located in the cell membrane, chloroplasts, and nucleus.

### 2.2. Chromosome Localization and Gene Duplication

Chromosomal localizations of *StGATA* genes were shown in Appendix A. *StGATA* genes numbered 1–13 were distributed in chr01, 14–18 in chr02, 19–20 in chr03, 21–27 in chr04, 28–33 in chr05, 34–38 in chr06, 39 in chr07, 40–45 in chr08, 46–48 in chr09, 49 in chr10, 50–51 in chr11, and 52–57 in chr12. Appendix A indicated 14 segmental duplications of *StGATA* genes. The pairs of segmental duplications were predicated including *StGATA12/17*, *StGATA12/14*, *StGATA5/25*, *StGATA3/46*, *StGATA14/17*, *StGATA15/18*, *StGATA17/19*, *StGATA14/19*, *StGATA20/43*, *StGATA31/55*, *StGATA34/40*, *StGATA34/44*, *StGATA38/50*, *StGATA39/52*, and *StGATA40/44*. To evaluate the positive selection and to speculate the possible date of duplicate events, the ratio Ka/Ks of the segmental duplications was calculated. These pair genes of segmental duplications may differentiate from the same ancestor gene during the evolutionary process to respond to natural selection, which may have occurred 14.95–66.48 million years ago (Appendix A).

### 2.3. Phylogenetic Analysis, Conserved Motifs, and Exon–Intron Organization of StGATA Family Members

To evaluate the evolutionary relationship of *StGATAs*, a polygenetic analysis was conducted using the amino acid sequences of 57 *GATA* genes from potato plants. As shown in Appendix A, *StGATAs* were clustered into four distinct groups: I, II, III, and IV. There were 20 *StGATAs* in subfamily I, 15 in subfamily II, 12 in subfamily III, and 10 in subfamily IV. Subfamily I comprised six groups, with two *StGATAs* in 1A, two in 1B, four in 1C, six in 1D, three in 1E, and three in 1F. Using the bioinformatics tool “MEME motif”, conserved motif domains among 57 *StGATA* genes were identified. This analysis was integrated with phylogenetic studies, revealing that the number of conserved motifs in each gene ranged from 1 to 14. The amino acid sequences of the conserved motifs are presented in Appendix A. Notably, the highest number of conserved domains was found in the gene *StGATA44*, which contained 14 motifs. The gene structure of *StGATAs* is closely related to the functional diversity and expression patterns observed among various members in potato plants. All *StGATAs* contain introns and exhibit a range of exon numbers from 1 to 11, as revealed by gene structure analysis. Notably, some *StGATAs* share the same number of exons and exhibit similar exon–intron patterns. Examples include *StGATA15/18*, *StGATA24/25*, *StGATA21/33*, *StGATA28/47*, and *StGATA36/37*. This structural similarity suggests potential functional redundancy among these genes. In contrast, the remaining *StGATA* genes display differences in the number of exons and distinct exon–intron patterns. We then analyzed the cis-regulatory elements in the promoter regions of *StGATA* genes. The detailed functions of the cis-regulatory elements are shown in Appendix A. Of note, these findings recommend that *StGATAs* are not only involved in hormonal signal transduction but also play a vital role under biotic and abiotic stress conditions and stimulate the growth, development, biochemical processes, physiological processes, and metabolism regulation in potato plants.

### 2.4. Syntenic Analysis of GATA Genes in 4 Plant Species

To analyze the similarity of gene arrangement in different genomes, synteny analysis was carried out between *Solanum tuberosum* (St), *Arabidopsis thaliana* (At), *Oryza sativa* (Os), and *Solanum lycopersicum* (Sl). Gene synteny also indicates the evolutionary relationship of genes in different species. In general, 26 potato *GATA* genes showed a closer syntenic relationship with those in *Solanum lycopersicum*, followed by *Arabidopsis thaliana* (21) and then *Oryza sativa* (5) (Appendix A).

### 2.5. Organ-Specific Expression of StGATA Genes and Expression Signatures of StGATAs in Potato Plants Under Biotic and Abiotic Stresses

Next, we detected the mRNA expression of *StGATA* genes in the potato (*Solanum tuberosum* L.) cultivar “Atlantic”. Figure 2A displays the expression of *StGATA* genes in six potato organs. mRNA expression of *StGATA* genes varied significantly in root, tuber, leaf, petiole, stem, and flower. Potato organs exhibited significant changes in mRNA expression of *StGATA* genes after treatment by NaCl (75 mM and 150 mM) (Figure 2B,C) and PEG6000 (10% and 20%) (Figure 2D,E). Notably, it was observed that *StGATA12* expression was significantly increased after treatment with NaCl (75 mM and 150 mM) and PEG6000 (10% and 20%). Consequently, *StGATA12* may be involved in the regulation of salt and osmotic stress in potato plant.

### 2.6. Phenotypic and Physiological Alterations Induced by StGATA12 Under NaCl and PEG Treatments

To confirm that *StGATA12* regulates the response of potato plant to salt stress and osmotic stress, *StGATA12*-overexpressing plants and RNA interference plants were generated and selected. The GFP-positive tobacco epidermal cells were observed under a laser scanning confocal microscope 48 h after p35S–StGATA12-GFP transfection. Figure 3A showed that chloroplasts emit red autofluorescence when excited. GFP is observable in the cytoblast, cytomembrane, and cytoplasm of plant cells. GFP-StGATA12 fusion proteins are observed in the nucleus. The relative expression level of *StGATA12* in sense lines (OE-1, OE-2, OE-3, OE-4, OE-5, OE-6, OE-7, and OE-8) was significantly increased with respect to non-transgenic plants (*p* < 0.05) (Figure 3B). In RNA interference plants (Ri-1, Ri-2, Ri-3, Ri-4, Ri-5, Ri-6, Ri-7, and Ri-8), expression of the endogenous *StGATA12* was down-regulated compared to NT plants (*p* < 0.05) (Figure 3C).

Under normal growth conditions, *StGATA12*-overexpressing and RNA interference plants showed no abnormal morphological phenotype compared with NT plants (Figure 4A). Salt stress and osmotic stress assays were carried out with these plants. Salinity and osmotic stress significantly decreased plant height, fresh plant weight, dry plant weight, fresh root weight, and dry root weight (Figure 4B–F). After NaCl treatment (75 mM and 150 mM), *StGATA12*-overexpressing plants were detected with increased plant height, fresh plant weight, dry plant weight, fresh root weight, and dry root weight compared to NT plants, while RNA interference plants showed the reverse trend. In response to 10% PEG treatment, *StGATA12*-overexpressing plants showed signs of increased growth, while RNA interference plants exhibited reduced growth compared to NT plants. However, there was no distinction in plant growth between the transgenic plants and non-transgenic plants after 20% PEG treatment. This may be due to the high concentration of PEG severely inhibiting the growth of potato tissue culture plantlets, resulting in no significant difference in growth between transgenic and non-transgenic plants.

Subsequently, we examined the effects of *StGATA12* on the accumulation of osmotic regulatory substances and the metabolism of reactive oxygen species under salt and osmotic stress conditions. The diverse physiological parameters, including levels of H_2_O_2_, MDA, proline, CAT, SOD, and POD, were assays for evaluating damages. Under normal conditions, *StGATA12*-overexpressing plants and RNA interference plants exhibited no significant changes in contents of H_2_O_2_ (Figure 5A,B), MDA (Figure 5C,D), and proline (Figure 5E,F), and showed no abnormal activities of CAT (Figure 5G,H), SOD (Figure 5I,J), and POD (Figure 5K,L). After 4 weeks of NaCl or PEG treatment, *StGATA12*-overexpressing plants had lower contents of H_2_O_2_ and MDA and higher proline levels, and elevated activities of CAT, SOD, and POD, compared with NT lines (*p* < 0.05). In contrast, RNA interference plants demonstrated higher contents of H_2_O_2_ and MDA, while decreased proline levels and activities of CAT, SOD, and POD were observed when compared with NT lines (*p* < 0.05).

### 2.7. Photosynthesis and Transpiration Affected by StGATA12 Under NaCl and PEG Treatments

On account of salinity and osmotic stress, the net photosynthetic rate (Figure 6A,B), transpiration rate (Figure 6C,D), and stomatal conductance (Figure 6E,F) were significantly decreased. The effect was more pronounced in *StGATA12*-overexpressing plants and RNA interference plants compared to the NT plants after NaCl or PEG treatment (*p* < 0.05). However, *StGATA12* overexpression improved net photosynthetic rate, transpiration rate, and stomatal conductance in response to salinity and osmotic stress. Under salinity and osmotic stress, the improvement of the photosynthetic capacity might be due to the closure of the stomata, which decreases the availability of internal CO_2_. *StGATA12* might be involved in this process in potato plant.

Chlorophyll is an important pigment involved in photosynthesis within plant cells, playing a crucial role in the process. This study further analyzed the changes in chlorophyll content in transgenic and non-transgenic potato plants under salt and osmotic stress conditions. At the beginning of the assays, NT, *StGATA12*-overexpressing, and RNA interference plants exhibited normal levels of chlorophyll content (Figure 7A,B). However, salt treatment and osmotic stress caused a constant and marked decrease in chlorophyll content for NT, *StGATA12*-overexpressing, and RNA interference plants. *StGATA12*-overexpressing plants have a higher content of chlorophyll than NT plants in response to salt stress or osmotic stress (*p* < 0.05). RNA interference plants lost more chlorophyll than NT plants (*p* < 0.05). Damage was evaluated by measuring ion leakage. In NT plants, ion leakage significantly increased after NaCl and PEG treatment (Figure 7C,D). In RNA interference plants, this parameter was obviously increased with respect to NT plants. *StGATA12*-overexpressing plants demonstrated decreased ion leakage relative to NT plants.

### 2.8. Effects of StGATA12 on Gene Expression in Response to Saline and Osmotic Stress

In order to avoid the production of reactive molecules, plants have evolved an effective scavenging system involving antioxidant enzymes. Superoxide dismutase (SOD, EC 1.115.1.1), catalase (CAT, EC 1.11.1.6), and peroxidase (POD, EC 1.11.1.7). SOD reacts with the superoxide radical to produce H_2_O_2_. Hydrogen peroxide is scavenged by CAT and POD. Saline and osmotic stress-induced significant increases in mRNA expression of *StSOD* (Figure 8A), *StCAT* (Figure 8B), and *StPOD* (Figure 8C) in the NT, *StGATA12*-overexpressing, and RNA interference plants. Our results have shown that under normal conditions, *StGATA12* overexpression enhanced the mRNA levels of *StSOD*, *StCAT*, and *StPOD*. Compared to the NT plants, there was a significant increase in mRNA levels of *StSOD*, *StCAT*, and *StPOD* in *StGATA12*-overexpressing plants, while these genes were down-regulated in RNA interference plants.

To counter the effects of saline and osmotic stress, plants have developed a number of mechanisms such as accumulation of compatible solutes, and proline is the most widely distributed amino acid as an osmotically active compound. Delta 1-pyrroline-5-carboxylate synthetase (P5CS, γ-glutamyl kinase, EC 2.7.2.11 and glutamate-5-semialdehyde dehydrogenase, EC 1.2.1.41) is a key regulatory enzyme involved in the biosynthesis of proline [28]. Treatment with NaCl or PEG resulted in a significant expression of the *StP5CS* gene in the NT plants (Figure 8D). After salinity and osmotic stress induction, *StP5CS* expression was significantly enhanced in *StGATA12*-overexpressing plants compared to the NT plants, while decreased in RNA interference plants. Interestingly, *StGATA12* overexpression per se induced the expression of *StP5CS*.

## 3. Discussion

The GATA transcription factor family plays diverse roles in plant growth, development, and responses to abiotic stresses. In this study, we identified 57 *StGATAs* in potato plants and classified them into four subfamily groups. The *StGATAs* exhibited organ-specific expression patterns and showed altered expression in response to salinity and osmotic stresses. Notably, *StGATA12* was found to modulate photosynthesis, transpiration, and stomatal conductance under these stress conditions. Furthermore, overexpression of *StGATA12* enhanced the biochemical responses of potato plants to salinity and osmotic stress by regulating the levels of H_2_O_2_, MDA, and proline, as well as the activities of CAT, SOD, and POD. It also induced genetic responses, including the expression of *StCAT*, *StSOD*, *StPOD*, and *StP5CS*, thereby enhancing the plant’s resilience to salinity and osmotic stress.

Previous studies have systematically identified multiple members of the *GATA* family genes in potato plants [29,30,31]. In this study, a total of 57 *GATA* genes were confirmed in potato plant. Based on the phylogenetic relationships, we defined four different subfamilies of *GATA* genes in potato plant, which is consistent with the classification reported by previous studies [29,30,31]. Further support for this classification comes from a comparison of motif constitutes and exon–intron structures, which are conserved among the members of each subfamily. The analysis of conserved motifs revealed that *StGATA12*, *StGATA13*, and *StGATA19* shared identical motifs, which were motifs 1, 2, 16, 23, and 26. This suggested a similarity in their molecular features and functions. The gene structure analysis indicated significant variation in the number of introns and exons among the *StGATA* genes.

To facilitate inferences of functional relationships, including potential redundancy between genes, we integrated expression pattern data for all 57 genes across six different samples. Strikingly, few *GATA* genes are expressed in specific organs neither under normal conditions nor under salt stress. Rather, *StGATA2* and *StGATA19* are highly expressed in flowers compared to other organs, revealing that the genes may be involved in flower development. A number of genes show increased expression after NaCl or PEG treatment, with the greatest differences observed for *StGATA12*. In plants, GATAs are known to be involved in the regulation of salt stress in Rice [26], common bean [32], grapevine [27], et al. For Solanaceae plants, *GATA* genes have been reported to respond to salt stress in wolfberry [33] and tomato [34]. In this study, we revealed the presence of 57 genes encoding the corresponding putative GATA transcription factors in the potato genome, which are differentially expressed in response to salinity stress.

Numerous studies have reported that GATAs regulate plant growth and development previously [24,35]. In poplar, *PdGATA19*-overexpressing transformants exhibited 25–30% faster growth, while CRISPR/Cas9-medicated mutant showed severely retarded growth compared with the wild type [24]. *GmGATA58* also alters the plant growth and inflorescence axis of Arabidopsis [35]. In this study, we found that the *StGATA12* gene also controls plant growth under normal conditions. Besides, in response to salinity stress, *StGATA12*-overexpressing plants show faster growth than the NT plants. However, the antisense lines exhibited the reverse effects on plant growth. Ion leakage was used to evaluate membrane permeability in the leaves. Salinity stress has been reported to disturb the integrity of cell membranes, resulting in increased membrane permeability [36]. *StGATA12* overexpression decreased ion leakage in response to salinity or osmotic stress. This result indicates that *StGATA12* may be helpful in osmotic adjustment or alleviating membrane permeability induced by salinity stress.

This study confirmed that *StGATA12* regulates net photosynthetic rate, transpiration rate, and stomatal conductance under salinity and osmotic stress. Many recent studies reinforce the perception that NaCl causes growth inhibition by changes in net photosynthetic rate, transpiration rate, and stomatal conductance [37,38,39]. Many studies have concluded that the reduction in photosynthesis in response to salinity is, to some extent, the result of reduced stomatal conductance [40,41]. Reduction in transpiration rate under salinity is further evidence of interference of salinity to stomatal conductance. Photosynthesis, a major contributor to crop yield, predominantly takes place in leaves where chlorophyll, one of the most abundant biological molecules in high plants, plays unique and essential roles in photosynthetic light harvesting and energy transduction. Reduction in photosynthesis activity under salt stress is due to the reduction in chlorophyll to a certain extent. Recently, *GATA* genes have been demonstrated to contribute to the regulatory mechanism of chlorophyll biosynthesis [35]. It is interesting that *StGATA12* overexpression did not lead to an increase in chlorophyll content under normal conditions, while in response to salinity stress, chlorophyll was increased in *StGATA12*-overexpressing plants. The reasonable explanation may be that *StGATA12* does not directly trigger the biosynthesis of chlorophyll but indirectly affects salinity or osmotic stress.

There is strong evidence that salinity-induced osmotic stress, ionic stress, and nutrient imbalance, as well as their secondary effects, altogether lead to the overexpression of reactive oxygen species [42]. SOD is the primary scavenger, which converts O_2_^−^ to H_2_O_2_. This toxic product of the SOD reaction is eliminated by catalase. Peroxidases are often the first enzymes that alter their activities under stress. Yu et al. reported that grape *GATA2* is involved in plant disease resistance by the H_2_O_2_ pathway and benefits transgenic *Arabidopsis* plants by preventing oxidative damage [43]. Here, the transgenic lines with significantly increased *StGATA12* gene expression exhibited reduced oxidative products, along with increased antioxidative components and higher contents of antioxidant enzymes, compared to the non-transgenic plants. Recent studies carried out in other plants have revealed the involvement of GATA transcription factors in the regulation of chlorophyll biosynthetic pathway genes under normal conditions [35]. Here, we found that *StGATA12* mediates the gene expression of the major components of the antioxidant defense system (*StSOD*, *StCAT*, *StPOD*, and *StP5CS*). This finding provided a base for further understanding the molecular evolution and functional characterization of the *StGATA12* gene in response to salinity and osmotic stress.

Conclusively, the *StGATA12* gene is highly expressed in potato leaves subjected to salinity and osmotic stress in a concentration-dependent manner. Plants overexpressing *GATA12* demonstrate normal growth, maintained photosynthetic capacity, and enhanced antioxidative ability when exposed to NaCl and PEG6000. Based on these findings, *StGATA12* can be identified as a candidate gene for salt tolerance, which could facilitate the development and utilization of saline land affected by seawater intrusion. However, future studies should focus more on assessing tuber yield and quality.

## 4. Materials and Methods

### 4.1. Identification of GATA Gene Family

Chromosome location, whole genome CDS, protein sequence, and genome length were downloaded from the Spud DB (http://spuddb.uga.edu/, accessed on 28 August 2022). Arabidopsis thaliana GATA protein sequence (TAIR10) was downloaded from the TAIR database (https://www.arabidopsis.org/Blast/index.jsp, accessed on 3 April 2022), and protein sequences with an e-value ≤ 1 × 10^−5^ were blasted. The hidden Markov model (HMM) of the GATA protein domain (PF00320) was obtained from the Pfam website [44]. The Hmmsearch procedure in HMMER Package Version 3.0 was used to search StGATA members from potato annotated protein sequences (DM v4.03/v4.04) with the default parameters [45]. GATA without conserved domains was removed after analysis of the assumed StGATA in the PfamScan database (https://www.ebi.ac.uk/Tools/pfa/pfamscan/, accessed on 3 April 2022) [46]. The repeated sequence was removed after the alignment using the MUSCLE algorithm in MEGA Version 11.0 (https://www.megasoftware.net/, accessed on 3 April 2022) [47,48].

### 4.2. Bioinformatic Prediction of Biochemical Properties and Subcellular Location

Amino acid composition, molecular weight (MV), isoelectric point (pI), and grand average of hydropathicity (GRAVY) of StGATA proteins were analyzed using ExPASy ProtParam (https://web.expasy.org/protparam/, accessed on 15 May 2022) [49]. Subcellular location was predicted using Plant-mPLoc (http://www.csbio.sjtu.edu.cn/bioinf/plant-multi/, accessed on 3 April 2023) [50] and ProComp 9.0 (http://www.softberry.com/berry.phtml?topic=protcomppl&group=programs&subgroup=proloc, accessed on 29 June 2022) [51].

### 4.3. Analysis of Chromosome Localization and Gene Duplication

All StGATA genes were localized onto potato chromosomes using Circos Version 0.69-9 (https://circos.ca/software/download/, accessed on 16 July 2022) based on physical location information obtained from the potato genome database Spud DB (http://spuddb.uga.edu/, accessed on 9 December 2022) [52]. The gene replication events of StGATA genes were analyzed using MCScanX (https://github.com/wyp1125/MCScanx, accessed on 3 April 2023) with the default parameters [53]. The synonymous substitution rate (Ks) and non-synonymous substitution rate (Ka) were calculated using TBtools (https://github.com/CJ-Chen/TBtools/releases, accessed on 23 April 2022). The evolutionary divergence times of StGATA genes were analyzed according to the method reported by Shen and Yuan [54].

### 4.4. Conserved Motif and Exon–Intron Structure

The conserved motifs of StGATA proteins were analyzed using MEME Software Version 5.3.0 (http://meme-suite.org/meme-software/5.3.0/meme-5.3.0.tar.gz, accessed on 8 October 2022) [55]. The parameters were set as follows: the number of motifs searched was 20, and the motif length range was 6–200 residues. All motifs were further annotated with InterProScan (http://www.ebi.ac.uk/interpro/, accessed on 23 April 2023) [56,57]. The CDS sequences and genomic sequences of 57 StGATA genes were downloaded from the Spud DB potato genome database. The CDS of the StGATAs were compared with their corresponding genomic DNA sequences using GSDS Version 2.0 (http://gsds.gao-lab.org/, accessed on 30 October 2023) to characterize the exon and intron distribution of *StGATAs* [58].

### 4.5. Phylogenetic Analysis and Classification of GATA Gene Family

The potato genome sequencing data were retrieved from the Spud DB database (DM v4.03/v4.04) [45,59], *Arabidopsis thaliana* genome sequences from The *Arabidopsis* Information Resource (TAIR10) (https://www.arabidopsis.org/, accessed on 21 January 2022), *Oryza sativa* v7.0 reference genome from the Rice Genome Annotation Project (http://rice.uga.edu/, accessed on 22 March 2022), and *Solanum lycopersicum* L. genome (ITAG Release 4.0) from the International Tomato Genome Sequencing Project (https://solgenomics.net/organism/Solanum_lycopersicum/genome/, accessed on 26 April 2022). Multiple alignment of protein sequences was performed using the MUSCLE algorithm [47]. Phylogenetic analysis was conducted using MEGA X software version 11.0 [48]. The p-distance was utilized to establish the neighbor-joining (NJ) tree with a 1000 replicate bootstrap.

### 4.6. Synteny and Collinearity Analysis

Makeblastdb was used to create a BLAST protein database [60]. BLASTP was used for the comparative analysis of GATA protein sequences [61]. Gene synteny and collinearity were analyzed using the MCScan X algorithm (https://github.com/wyp1125/MCScanx, accessed on 3 April 2023) [53].

### 4.7. Promoter Sequence Analysis

Custom perl scripts were used to retrieve the DNA sequence from +2000 bp upstream of the transcription start codon (ATG) in the 5′-UTR. Promoter sequences were identified through the PlantCARE database (http://bioinformatics.psb.ugent.be/webtools/plantcare/html/, accessed on 13 May 2022) [62].

### 4.8. Plant Material and Treatment

Four-week-old, virus-free tissue culture seedlings of the Atlantic potato variety that were in similar growth conditions were selected. After removing the top and bottom parts, the middle section containing two axillary buds was cut into 2 cm long segments. The segments were inoculated into 150 mL Erlenmeyer flasks containing 30 mL of MS liquid medium (without agar, containing 30 g/L sucrose, pH 5.8 ± 0.1). Each flask contained 3 stem segments and was cultured under conditions of 21 °C temperature and 2800 lx light intensity with a 16 h/day photoperiod for 4 weeks.

To induce tuber production, 4-week-old seedlings were cultured in an MS medium containing 8% sucrose. The sprouted tubers were grown in pots (20 cm diameter × 40 cm height) containing a 1:1 (*v*/*v*) mixture of nutrient soil and perlite, with soil moisture maintained at 70–75%. The sprouted tubers were cultivated until the flowering stage to collect flowers, leaves, stems, petioles, and roots, and were then grown to maturity for tuber collection. Samples were frozen in liquid nitrogen and stored at −80 °C for quantitative reverse transcription PCR (RT-qPCR) analysis of *StGATAs*. To assay the expression patterns of *StGATAs* in response to salinity and osmotic stress, the well-growing 4-week-old seedlings were selected and inoculated into 30 mL of liquid medium containing 0%, 10%, and 20% PEG6000 (W/V), as well as 0, 75, and 150 mM NaCl. After culturing for 0, 1, 3, 6, 12, and 24 h, the leaves were collected for the examination of *StGATAs* mRNA expression. After 24 h of treatment, the leaves were collected for evaluation of physiological parameters, photosynthetic gas exchange parameters, enzyme activities, chlorophyll contents, and gene expression. The leaves collected were the fully expanded third leaf from the top of the plant, gathered between 9:30 and 11:30 AM. To evaluate phenotypes and plant growth, 4-week-old seedlings were cultured for 6 days, followed by transferred into a medium containing PEG or NaCl. After 4 weeks, we analyzed plant phenotypes and measured plant height, fresh plant weight, dry plant weight, fresh root weight, and dry root weight.

### 4.9. Plasmid Construction and Transfection

To create *StGATA12*-overexpressing plants (referred to as OE), the gene encoding the StGATA12 protein was cloned using the following primers: forward 5′-CTCGAGATGTCTATGAAAAATACCCAAC-3′ and reverse 5′-GTCGACACAAGTTGAAATCATAGAAGCTAA-3′ into the pBI121-EGFP plasmid according to a previously reported approach [63]. An RNA hairpin was created by cloning the *StGATA12* cDNA fragment (designated as Ri) using the following primers: forward 5′-TGTCTATGAAAAATACCCAACA-3′ and reverse 5′-ATCCACAAATTGGGATAACCATT-3′ into the gene-silencing vector pHannibal [64]. Agrobacterium containing the plasmids was cultured for approximately 48 h in LB medium supplemented with 50 mg/L gentamicin and 50 mg/L spectinomycin at 28 °C. The culture was then harvested by centrifugation (5000 rpm for 10 min) and re-suspended in MS medium to an optical density of 0.3 (OD600). The sterile seedling stems (2 cm) were incubated in the Agrobacterium suspension for 10 min, then transferred to MS medium (pH 5.8) containing 7.4 g/L agar, 30 g/L sucrose, 0.5 mg/L 6-BA, 2.0 mg/L ZT, 0.2 mg/L GA3, and 1.0 mg/L IAA. The samples were maintained in the dark for 48 to 72 h. Next, the plants were transferred to differentiation media consisting of MS, 7.4 g/L agar, 30 g/L sucrose, 300 mg/L Timentin, 100 mg/L kanamycin, 0.5 mg/L 6-BA, 2.0 mg/L ZT, 0.2 mg/L GA3, and 1.0 mg/L IAA (pH 5.8). The media were changed every 2 weeks. After the induction of adventitious buds, the resistant buds were transferred to a rooting medium composed of MS, 7.4 g/L agar, 30 g/L sucrose, 300 mg/L Timentin, and 100 mg/L kanamycin (pH 5.8) until adventitious roots were induced.

### 4.10. Subcellular Location

The constructs were cloned with the following primers: forward 5′-CTCGAGATGGATTACTCCGGCAACTGTCAAAGT-3′ and reverse 5′-GTCGACAAAACTCTGAACCGGCGGACCCGGTTCAGT-3′. They were then fused into pCAM35s-GFP, which was then transformed into *Agrobacterium tumefaciens* GV3101. The transformed strain was utilized to infiltrate tobacco epidermal cells according to the method reported by Sparkes et al. [65]. Infected areas were marked, and green fluorescence was detected using a Leica TCA confocal scanning laser microscope (Leica, Wetzlar, Germany) 48 h after infiltration. Green fluorescence was detected after emission/excitation at 510/488 nm, and chloroplast fluorescence was examined after emission/excitation at 675/640 nm.

### 4.11. Physiological Evaluation

#### 4.11.1. Ion Leakage

The fully expanded top-third functional leaves were collected from transgenic and non-transgenic potato plants at the same position. The leaves were washed twice with distilled water to remove any adhering electrolytes from the surface or cut ends of the tissues. Surface water was absorbed using filter paper. Leaf discs were punched from the potato leaves using a 10 mm punch, ensuring to avoid the central veins. These discs were placed in a 0.1 mol/L mannitol aqueous solution and shaken at 100 rpm for 2 h. Relative electrolyte leakage was measured using a conductometer (DDS-11A, Shanghai Scientific Instruments, Shanghai, China). The initial conductivity of the solution was recorded as L1. The solution was then boiled for 10 min, cooled to 20 °C, and the final conductivity was measured as L2. Ion leakage was expressed as the ratio of L1 to L2.

#### 4.11.2. H_2_O_2_ Content

H_2_O_2_ content was determined using a previously described method with minor modifications [66]. Briefly, 0.5 g of leaves was extracted with 5 mL of TCA (0.1%, *w*/*v*), and the extract was centrifuged at 12,000 rpm for 15 min. The supernatant (0.5 mL) was then collected and diluted with 1 mol/L KI and 0.5 mL of potassium phosphate buffer (10 mM, pH 7.0). The absorbance was measured spectrophotometrically at 390 nm.

#### 4.11.3. MDA Content

The MDA content was measured using a modified version of Heath’s method [67]. Briefly, 0.2 g of fresh leaves was extracted with 5 mL of TCA (10%). The extract was then centrifuged at 4000× *g* for 10 min, and the supernatant was collected. A 2 mL aliquot of the supernatant was mixed with 2 mL of 0.6% TBA prepared in 10% TCA and incubated at 100 °C for 15 min. After centrifugation at 3500 rpm for 10 min, the absorbance was measured at 532 nm, 600 nm, and 450 nm.

#### 4.11.4. Proline Content

Proline content was measured using a modified version of Bates’ method [68]. Briefly, 0.2 g of potato leaves was homogenized in 5 mL of 3% sulfosalicylic acid and the mixture was heated in a boiling water bath for 10 min. After cooling, 2 mL of the supernatant was combined with 3 mL of 2.5% ninhydrin and 2 mL of acetic acid. The color reaction was allowed to proceed for 40 min in a boiling water bath. The product was then extracted with toluene, and the absorbance was measured at 520 nm.

### 4.12. Photosynthetic Gas Exchange Parameters

The third leaf from the top of the plant was collected when fully expanded between 9:30 and 11:30 AM. The net photosynthetic rate, transpiration rate, and stomatal conductance were measured using a portable LI-6400XT photosynthesis system (Li-COR, Lincoln, NE, USA). The photon flux density was set to 1500 μmol·m^−2^·s^−1^, with relative humidity in the leaf chamber maintained at 50% to 70%. The CO_2_ concentration was set at 400 μmol/mol.

### 4.13. Activity of Catalase (CAT), Superoxide Dismutase (SOD), and Peroxidase (POD)

#### 4.13.1. CAT Activity

To measure CAT activity, 2.5 g of leaves were homogenized in 25 mL of phosphate-buffered saline (pH 7.8). The mixture was centrifuged at 4000 rpm for 15 min to collect the supernatant. This supernatant was then incubated with 2.5 mL of 0.1 mol/L H_2_O_2_ at 30 °C for 10 min. The reaction was terminated by adding 2.5 mL of 10% H_2_SO_4_. CAT content was determined by titration with 0.1 M KMnO_4_ in the presence of H_2_SO_4_. An extracted solution that had been boiled for 5 min served as a control.

#### 4.13.2. SOD Activity

To measure SOD activity, 0.5 g of leaves were homogenized in phosphate-buffered saline to obtain a 5 mL mixture. The supernatant was collected by centrifugation at 1000 rpm for 20 min. A 0.05 mL aliquot of the extract was incubated with a chromogenic reagent composed of 1.5 mL of 0.05 mol/L phosphate-buffered saline, 0.3 mL of 130 mM methionine, 0.3 mL of 750 μM nitroblue tetrazolium, 0.3 mL of 100 μM EDTA-Na_2_, 0.3 mL of 20 μM riboflavin, and 0.25 mL of H_2_O under 4000 Lux for 20 min. A separate mixture was maintained in the dark as a control. The absorbance was measured at 560 nm.

#### 4.13.3. POD Activity

To measure POD activity, 5.0 g of leaves were homogenized in 10 mL of phosphate-buffered saline. The supernatant was collected by centrifugation at 3000× *g* for 10 min and transferred to a 25 mL volumetric flask, then diluted with phosphate-buffered saline. A 0.1 mL aliquot of the extracted solution was incubated with a reaction mixture containing 2.9 mL of 0.05 mol/L phosphate-buffered saline, 1.0 mL of 2% H_2_O_2_, and 1.0 mL of 0.05 mol/L guaiacol at 37 °C for 15 min. The reaction was terminated by adding 2.0 mL of TCA. The reaction mixture was then filtered, and the absorbance was measured at 470 nm. An extracted solution that had been boiled for 5 min served as a control.

### 4.14. Chlorophyll Content

Chlorophyll content in potato leaves was measured using a commercial chlorophyll assay kit according to the manufacturer’s instructions (Solarbio, Beijing, China). Fresh potato leaves were collected and washed with distilled water. After draining the surface moisture, the midrib was removed, and the leaves were cut into pieces. Approximately 0.1 g of the leaves was weighed and ground thoroughly in 1 mL of water in the dark. The mixture was then transferred into a 10 mL volumetric flask, diluted with water to the mark, and mixed thoroughly. The volumetric flask was kept in the dark for 3 h. The absorbance of the supernatant was measured at wavelengths of 663 nm and 645 nm using a spectrophotometer (model 552, Perkin Elmer, Shelton, CT, USA).

### 4.15. qRT-PCR

Relative mRNA expression was assessed using qRT-PCR. cDNA synthesized from total RNA extracted from plant leaves served as the template for analysis. The qPCR reaction mixture contained 100 ng of cDNA, 0.6 µL of specific primers (10 µM), 10 µL of 2 × SuperReal PreMix Plus, and 0.4 µL of 50 × ROX Reference Dye (Tiangen Biotech, Beijing, China), adjusted to a final volume of 20 µL. The thermal cycling conditions on the ABI 3000 system (Applied Biosystems, Foster City, CA, USA) were as follows: initial denaturation at 94 °C for 2 min, followed by 40 cycles of denaturation at 94 °C for 30 s, annealing at 60 °C for 34 s, and extension at 72 °C for 30 s. Cycle threshold (CT) values were recorded, and mRNA levels were calculated using the formula 2^−ΔΔCt^ [69]. Each experiment comprised three technical replicates and three biological replicates, with *StEf1a* serving as an internal control. Specific primers used in this study are listed in Table 1.

### 4.16. Statistical Analysis

Statistical analysis was conducted using GraphPad Prism Software Version 9 (GraphPad, San Diego, CA, USA) and IBM SPSS Statistical Software Version 19 (IBM, Chicago, IL, USA). The Kolmogorov–Smirnov test and the Shapiro–Wilk test were employed to assess the normality of the data. Levene’s test was used to evaluate homoscedasticity. All data were found to fit a normal distribution and exhibited homoscedasticity. Results are presented as the mean ± standard deviation. The statistical significance of differences in mean values was determined using one-way ANOVA, with corrections applied using Dunnett’s method.

## Figures and Tables

**Figure 1 ijms-25-12423-f001:**
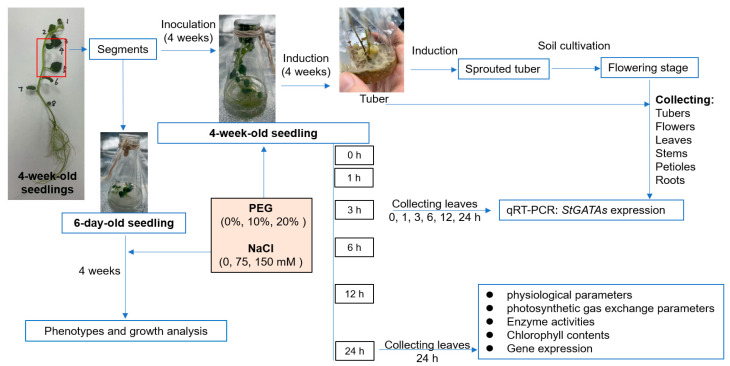
Study design flowchart.

**Figure 2 ijms-25-12423-f002:**
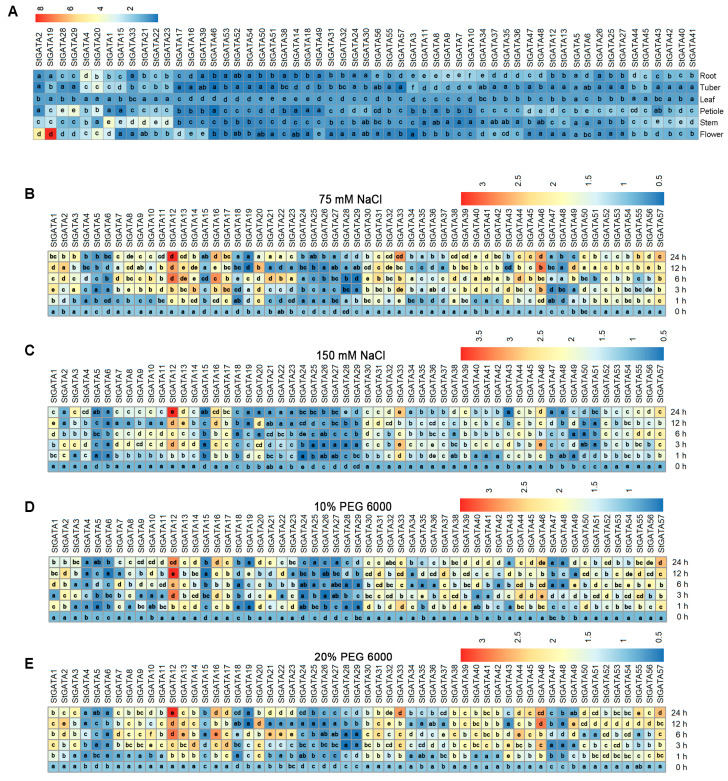
Heatmap presenting mRNA expression of *StGATA* family genes in different potato plant organs and in leaves responding to NaCl-induced salt stress and PEG-induced osmotic stress. (**A**) mRNA expression of *StGATA* genes in potato roots, tubers, leaves, petioles, stems, and flowers; different letters indicate significant difference (*p* < 0.05, by one-way ANOVA with Tukey test or Dunnett’s T3 for post hoc analysis) among root, tuber, leaf, petiole, stem, and flower. mRNA expression profiles of *StGATA* genes under (**B**,**C**) salt stress and (**D**,**E**) osmotic stress; different letters indicate significant difference (*n* = 9, *p* < 0.05, by one-way ANOVA with Tukey test or Dunnett’s T3 for post hoc analysis) among leaf samples; four-week-old normally grown plants were subjected to 0 h, 1 h, 3 h, 6 h, 12 h, and 24 h of cultivation with NaCl (75 mM and 150 mM) or PEG6000 (10% and 20%).

**Figure 3 ijms-25-12423-f003:**
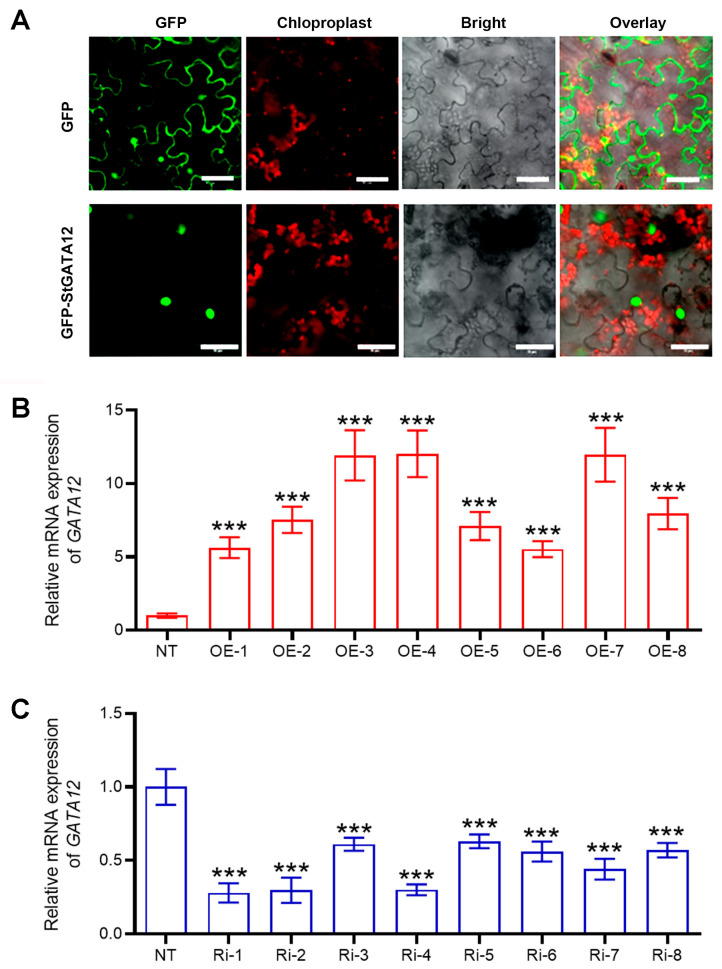
Subcellular localization of GATA12 and mRNA expression of *GATA12* gene in the transgenic plants. (**A**) StGATA12 protein located on the cellular nucleus of tobacco epidermal cells; GFP-StGATA12 fusion protein was transiently expressed in tobacco leaves and observed using a laser scanning confocal microscope; Scale bar = 50 μm. (**B**,**C**) The relative quantification of *StGATA12* mRNA in pBI121-EGFP-StGATA12-transgenic lines and pART-StGATA12-RNAi-transgenic lines; Data are the means ± standard deviation. *** *p* < 0.001 (OE or Ri compared to NC, two-way ANOVA corrected by Sidak’s multiple comparisons test).

**Figure 4 ijms-25-12423-f004:**
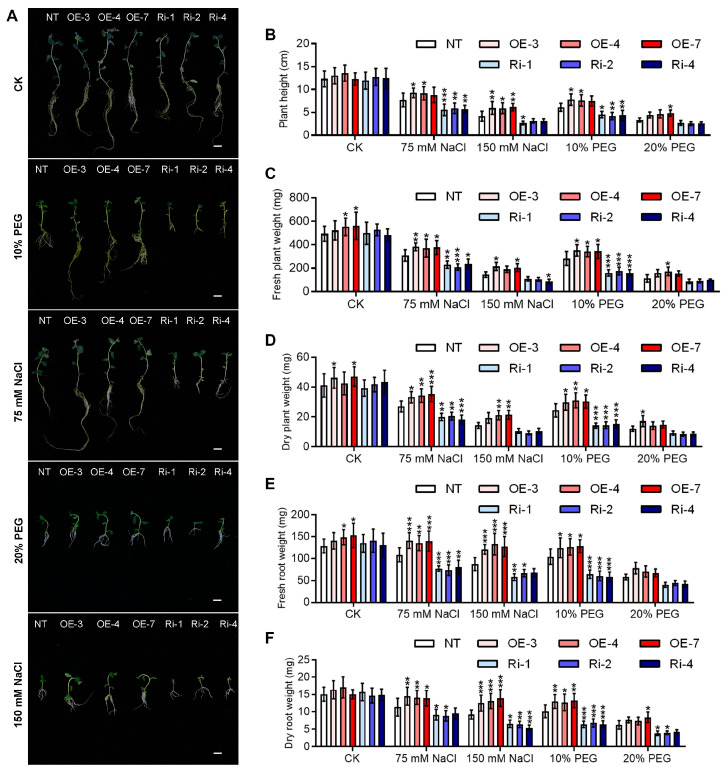
Phenotypical and growth alterations of non-transgenic and transgenic lines in response to NaCl-induced salt stress and PEG-induced osmotic stress. (**A**) Representative phenotypes of NT and transgenic plants; bar = 2 cm. Morphological changes ((**B**), plant height; (**C**), fresh plant weight; (**D**), dry plant weight; (**E**), fresh root weight; (**F**), dry root weight) of potato plants were imaged 2 days after cultivation with NaCl (75 mM and 150 mM) or PEG6000 (10% and 20%). NT, non-transgenic plants; OE, pBI121-EGFP-StGATA12-transgenic lines; Ri, pART-StGATA12-RNAi-transgenic lines. Mean ± standard deviation (*n* = 9). Ordinary two-way ANOVA with Tukey’s multiple comparisons test, * *p* < 0.05, ** *p* < 0.01, *** *p* < 0.001.

**Figure 5 ijms-25-12423-f005:**
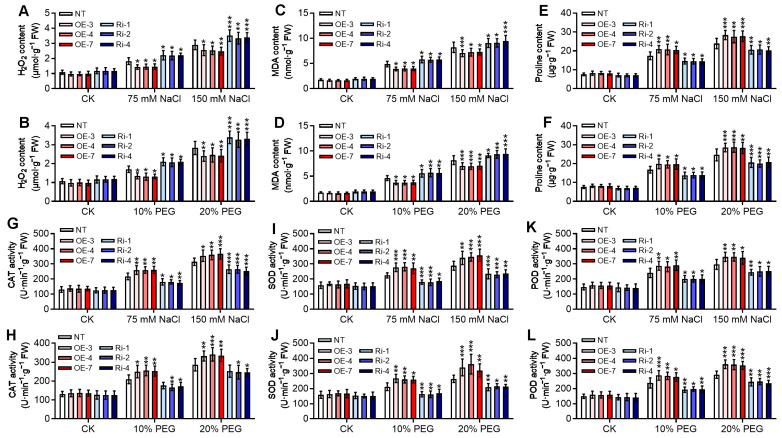
Physiological indexes of non-transgenic and transgenic plants in response to NaCl-induced salt stress and PEG-induced osmotic stress. (**A**,**B**) H_2_O_2_ content, (**C**,**D**) MDA content, (**E**,**F**) proline content, (**G**,**H**) CAT activity, (**I**,**J**) SOD activity, and (**K**,**L**) POD activity were examined 24 h after cultivation with NaCl (75 mM and 150 mM) or PEG6000 (10% and 20%) treatment. NT, non-transgenic plants; OE, pBI121-EGFP-StGATA12-transgenic lines; Ri, pART-StGATA12-RNAi-transgenic lines. Mean ± standard deviation (*n* = 9). Ordinary two-way ANOVA with Tukey’s multiple comparisons test, * *p* < 0.05, ** *p* < 0.01, *** *p* < 0.001.

**Figure 6 ijms-25-12423-f006:**
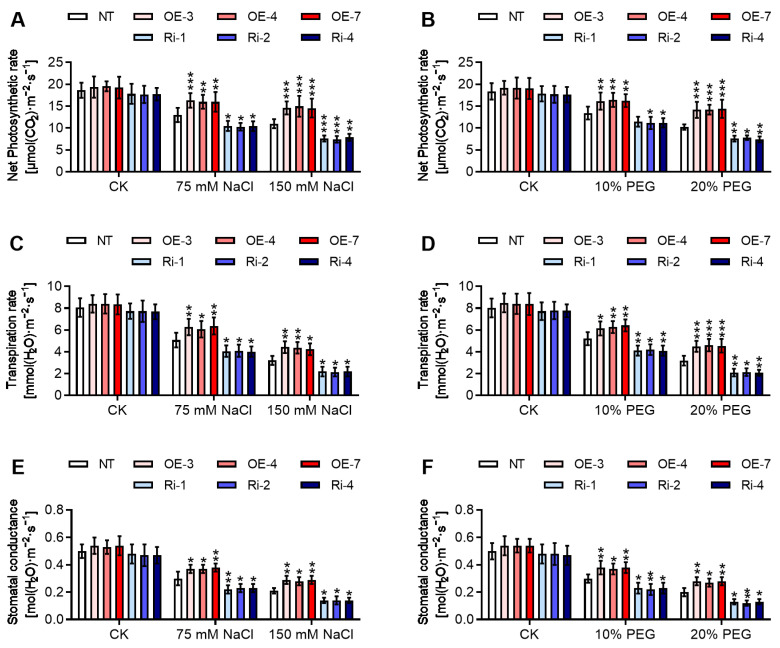
Photosynthesis of non-transgenic and transgenic plants in response to NaCl-induced salt stress and PEG-induced osmotic stress. (**A**,**B**) Net photosynthesis rate, (**C**,**D**) transpiration rate, and (**E**,**F**) stomatal conductance were examined 24 h after cultivation with NaCl (75 mM and 150 mM) or PEG6000 (10% and 20%). NT, non-transgenic plants; OE, pBI121-EGFP-StGATA12-transgenic lines; Ri, pART-StGATA12-RNAi-transgenic lines. Mean ± standard deviation (*n* = 9). Ordinary two-way ANOVA with Tukey’s multiple comparisons test, * *p* < 0.05, ** *p* < 0.01, *** *p* < 0.001.

**Figure 7 ijms-25-12423-f007:**
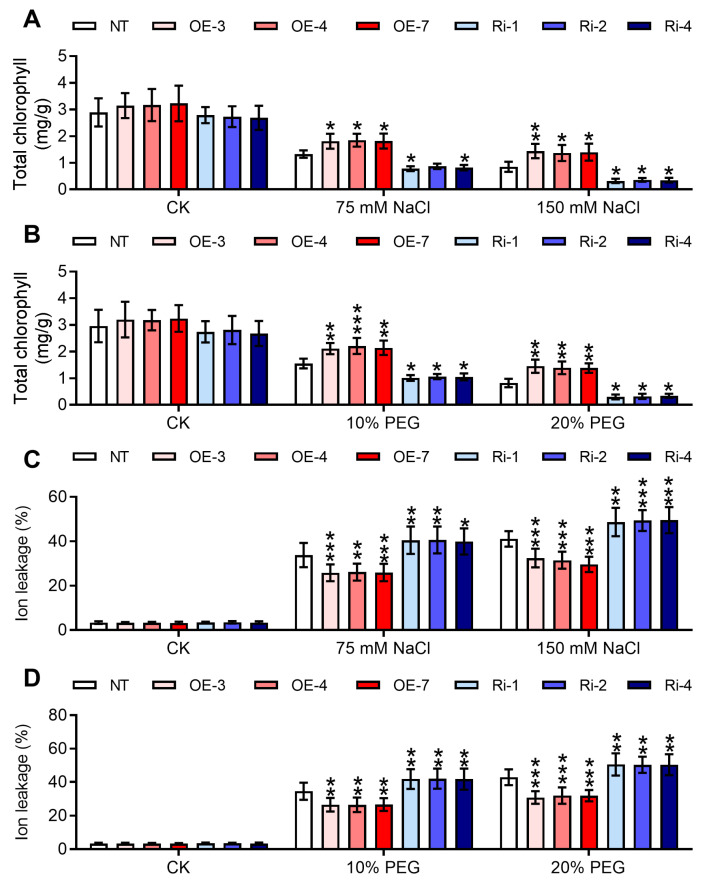
Chlorophyll content and ion leakage of non-transgenic and transgenic plants in response to NaCl-induced salt stress and PEG-induced osmotic stress. Chlorophyll content (**A**,**B**) and ion leakage (**C**,**D**) were analyzed 24 h after cultivation with NaCl (75 mM and 150 mM) or PEG6000 (10% and 20%). NT, non-transgenic plants; OE, pBI121-EGFP-StGATA12-transgenic lines; Ri, pART-StGATA12-RNAi-transgenic lines. Mean ± standard deviation (*n* = 9). Ordinary two-way ANOVA with Tukey’s multiple comparisons test, * *p* < 0.05, ** *p* < 0.01, *** *p* < 0.001.

**Figure 8 ijms-25-12423-f008:**
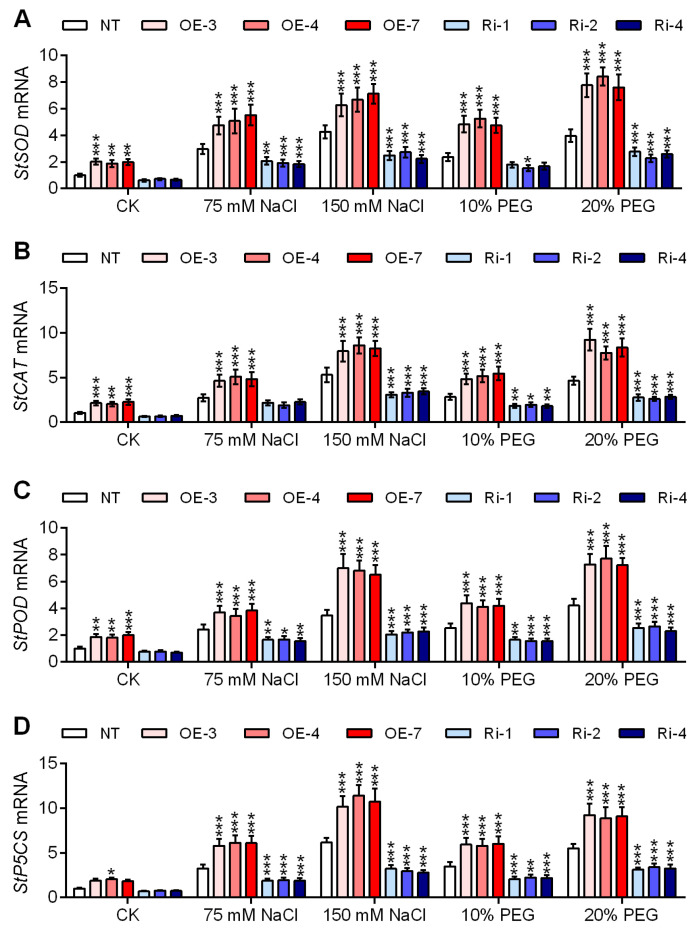
mRNA expression of stress-responsive genes in non-transgenic and transgenic plants in response to NaCl-induced salt stress and PEG-induced osmotic stress. (**A**) *StSOD* mRNA, (**B**) *StCAT* mRNA, (**C**) *StPOD* mRNA, and (**D**) *StP5CS* mRNA in the leaves were assayed 24 h after cultivation with NaCl (75 mM and 150 mM) or PEG6000 (10% and 20%). NT, non-transgenic plants; OE, pBI121-EGFP-StGATA12-transgenic lines; Ri, pART-StGATA12-RNAi-transgenic lines. Mean ± standard deviation (*n* = 9). Ordinary two-way ANOVA with Tukey’s multiple comparisons test, * *p* < 0.05, ** *p* < 0.01, *** *p* < 0.001.

**Table 1 ijms-25-12423-t001:** Specific primers used in this study.

Gene ID	Gene	Forward (5′-3′)	Reverse (5′-3′)	Length (bp)
XM_006347752.2	*StEf1α*	GGTTGTATCTCTTCCGATAAAGGC	GGTTGTATCTCTTCCGATAAAGGC	132
XM_015308529.1	*StP5CS*	TGCAATGCAATGGAAACGCT	ACAATTTCCACGGTGCAAGC	194
AY442179	*StCAT*	CCATGCTGAGGTGTATCCTATTC	CCTTTCTCCTGGTTGCTTGA	100
AF354748	*StSOD*	CATTGGAAGAGCTGTTGTTGTT	ATCCTTCCGCCAGCATTT	96
XM_006362636.2	*StPOD*	AGATGTTGTGGCCATGTCTGG	GCTTGTGTTGAAGGATGGAGC	118
PGSC0003DMT400006011	*StGATA1*	TGGAGATCGCAACTCCAGAAG	GCATTGCACAGCGACTTAGG	196
PGSC0003DMT400066953	*StGATA2*	CTCTGCGTTCCCAGTGATGA	TTCCGTGAAACGACGCAGTA	136
PGSC0003DMT400063771	*StGATA3*	TCCGACCCAAAAGGAGGAATC	TCCCACAAGCATTGCACAGT	137
PGSC0003DMT400061125	*StGATA4*	ATTGTGGCATCAAGCAAACCG	CTGTTTGCATGTCGGGGAAC	130
PGSC0003DMT400083773	*StGATA5*	CACCAAGATAGACACCAGCAAAC	TAGCGAATGTGAGGAGTAGGGTT	136
PGSC0003DMT400083774	*StGATA6*	GTGGAAGTGGAGGAGGAATAGAG	CCACACCATTCATCTCCATAGAAC	131
PGSC0003DMT400027729	*StGATA7*	CCCTGTAGATAGCGGCAGAGTTA	CCTCACACCCTCCTAATAATAGCA	142
PGSC0003DMT400027731	*StGATA8*	TGTGGGCTTATGTGGGCAAA	AGCTTCTTGCATATCCTCTTGAT	165
PGSC0003DMT400027733	*StGATA9*	CAGTTAACATTGATGAAGAGCAGGA	GCAGGCCAGTGAACACTCAT	130
PGSC0003DMT400027730	*StGATA10*	ACCACTGCTGAGCTGGATATG	TCCTGCTCTTCATCAATGTTAACTG	101
PGSC0003DMT400027728	*StGATA11*	ATGTGGGCAAACAAGGGTAT	CAGAAAGGAGAACTATCAGCAAAGT	187
PGSC0003DMT400008112	*StGATA12*	CTTCTTCTCCAACCTCTTCGTC	GGTCTCTGCTGATGGATTCTTT	142
PGSC0003DMT400008111	*StGATA13*	ACTCCCTATTCCGGTTGATGA	CGGTTCAGACCGGAACTCTG	145
PGSC0003DMT400031356	*StGATA14*	GAACTCAGTCTTCCTGGGGC	TCGTTCGGCACTAAACGGAA	192
PGSC0003DMT400040074	*StGATA15*	TTCTTCTTCGTCTTCCGTTGAT	ATGTAAGAAGAAGAAGAGCGACG	119
PGSC0003DMT400009117	*StGATA16*	AGTGAAATCAGTGTTCCGACTG	CCGGCAGGATAAGCAAGTGA	101
PGSC0003DMT400009118	*StGATA17*	CTTGTGGTGTTCGGTTCAAATC	TTTCTTCTGATTCCTTCTTCCG	137
PGSC0003DMT400009031	*StGATA18*	CATCGTTGTAGTGGGAGTATGGT	TGATAAGGCGAGTAGAAGGAGTTC	189
PGSC0003DMT400089018	*StGATA19*	CAAATTTCACCGTCTCTCTCACA	GAAGCTGTCCATCCCCTGC	100
PGSC0003DMT400006491	*StGATA20*	CGAACTCTGCGTTCCGTTTG	TGAACTGTCGGTGGTGATGG	148
PGSC0003DMT400070134	*StGATA21*	ATTCCTGACCTCAAGTCCTGTTT	AGAAACAGGACTTGAGGTCAGGA	124
PGSC0003DMT400070133	*StGATA22*	TTGGAACGATCCGTTGCCTG	AACGATATCCTCGTACGGAACT	106
PGSC0003DMT400070135	*StGATA23*	TCGGATTTCGTGGATGAGATAG	TCCTTACAATCAACAGCGTCAA	112
PGSC0003DMT400024208	*StGATA24*	TTGTGCTGATCTGGAAAAGAATC	TGCAGGAATGACGACCTCAG	154
PGSC0003DMT400024207	*StGATA25*	ATAGTGTCAGGAAAGAGGTTGCT	ACAGTTACAGAATGTTGTGTGCC	141
PGSC0003DMT400024206	*StGATA26*	GATGGAGGAGAAGAGACTATGGAT	TAGAAACTTCCACAACTCCACCT	123
PGSC0003DMT400024205	*StGATA27*	GGGAACTCCTGACAATCCCG	ACAGGCAGTCAACCTCAGTT	203
PGSC0003DMT400069864	*StGATA28*	AGCCCTTCATTTCCTGATTATGT	ACTGAATTTGGGCTGTGGTGA	137
PGSC0003DMT400069865	*StGATA29*	GCCCTTCATTTCCTGATTATGT	GTTGTTGTTGTTGTTGTTGCTG	172
PGSC0003DMT400060240	*StGATA30*	CAGCAGCAACAGTGAAGATAGTAA	AATGCTGCTTGTTCTACTTCTCC	167
PGSC0003DMT400060241	*StGATA31*	GGTGGATCTAAGTGATAAACAGGGT	GCAGGTCCACCTCTCCAAAG	141
PGSC0003DMT400060242	*StGATA32*	GCAGCAGCAACAGTGAAGATAGTA	ATGCTGCTTGTTCTACTTCTCCAA	167
PGSC0003DMT400059990	*StGATA33*	AGCTCTCAGTTCCGTATGAGG	GCCTTGGGATAACGGCTCTT	134
PGSC0003DMT400068348	*StGATA34*	ACAGCCTTCTCAAGGACACA	GCTTTCTTTGCACCTGCATACT	148
PGSC0003DMT400068347	*StGATA35*	GACATCCGAACTCAATAGGTAGAG	GTAGACAATCGTGAATAAGCCTCA	102
PGSC0003DMT400068346	*StGATA36*	AGTGACAAGCCTATGGTCTCTGTT	GGTAGACAATCGTGAATAAGCCTC	155
PGSC0003DMT400068349	*StGATA37*	GAAGTTACAGGAGGGCCCAA	CTGAAGCGAAATGGTCTGCAT	103
PGSC0003DMT400062488	*StGATA38*	CGAGGAAGATTGGGATGCGA	GGGACTCCAGAAATTCGTTAGGA	146
PGSC0003DMT400011449	*StGATA39*	TGGGAGATCAAAAGCAACAACC	CCACATGCGTTACACAATGACTTA	145
PGSC0003DMT400052800	*StGATA40*	TCCGAAGTGTTCAGGTGCAA	AGGCAAACGAGCTTCTTGGA	128
PGSC0003DMT400052799	*StGATA41*	AGAACCTTGTGACTTTGAGGAACA	GAGCCAGAACTTGACCTATTGCTA	188
PGSC0003DMT400052801	*StGATA42*	TCCGAAGTGTTCAGGTGCAAA	TTGGACTGAAGCAAGAGCCAT	113
PGSC0003DMT400074935	*StGATA43*	TTGTCCGGAAGCAATCACCC	CAGCCATATCTTCATATGGAACGG	100
PGSC0003DMT400067506	*StGATA44*	GTCGGTTGACAACAAGCACC	GGTGGTCCCTGCTCCTTTTA	170
PGSC0003DMT400067505	*StGATA45*	ACAACAATGCTCATACTTCTCTGG	GGCTTCTGATTCTTCTTCTCTACC	144
PGSC0003DMT400081417	*StGATA46*	CATCAGGTCCCAAGTCGTTG	GCCATCAATAATATCGCCGCT	187
PGSC0003DMT400030274	*StGATA47*	CAACTGCATGTTTCATGGTGGA	TTCTTCCTCTACACACTCAGGG	179
PGSC0003DMT400030276	*StGATA48*	GTGGATGATGACCTTCTCAACTTC	GAAGAAGGCTAACAAGAGGGTTTG	135
PGSC0003DMT400030708	*StGATA49*	ACCAACCACCTCCTACCGAT	TGCTACATCATCACTCGGAACA	110
PGSC0003DMT400020876	*StGATA50*	CATCCACACCCTCCGATCAA	CGAGGACGTACGGGAATGAC	152
PGSC0003DMT400020875	*StGATA51*	ACATCCACACCCTCCGATCA	ACTCTTTCCACCAGAGCAGG	112
PGSC0003DMT400000761	*StGATA52*	AGCAACAGCTCTTCCAACAAC	CATGCGTTACAAAGAGACTTAGGG	119
PGSC0003DMT400000760	*StGATA53*	CTCAAACTCTCACAGGAAAGTCGT	TACTCATAGGAACAAACTCTGGCG	172
PGSC0003DMT400000762	*StGATA54*	CTGATTACAGCAGCAACAGCTC	CCACATGCGTTACAAAGAGACTTA	133
PGSC0003DMT400011779	*StGATA55*	GACCTGCTGGACCTAAGTCAT	TTTCTCCGCTGCTGCTTGTA	186
PGSC0003DMT400011778	*StGATA56*	ATGTGGAATAAGGAGCAGGAAGA	GCTACTCTGGTTCTGAGGATGATG	120
PGSC0003DMT400011780	*StGATA57*	ACCTGCTGGACCTAAGTCATTGT	TGCTACTGCTATTGCTACTCTGGTT	164

## Data Availability

The original contributions presented in this study are included in the article/Appendix A. Further inquiries can be directed to the corresponding authors. The GATA protein sequence described in this article can be obtained from the Potato genome resources using the DM v4.03 database (https://spuddb.uga.edu/integrated_searches.shtml, accessed on 24 August 2023).

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
