# Peer review of "Genome-Wide Identification of *GATA* Family Genes in Potato and Characterization of *StGATA12* in Response to Salinity and Osmotic Stress"

_ijms, 2024, doi:10.3390/ijms252212423_

Round 1

Reviewer 1 Report

Comments and Suggestions for Authors

The production of potato is threatened by salinity. Potato plant growth is inhibited under salt stress due to ion toxicity caused by osmotic stress. This study identified the GATAs in potato and investigates the role of StGATA12, in response to salinity and osmotic stress. The findings indicate that StGATA12 enhances the growth and photosynthetic performance of potato plants under stress conditions by regulating key physiological and biochemical responses, suggesting its potential for improving the plant's resilience to adverse environmental conditions. These findings are significant for improving potato yield and quality. Here are some comments about this article.

Firstly, some experiments need to be improvement. In Figure 5, the data presented maybe from the in vitro seedlings. How about the phenotypes of these transgenic plants in the growth chamber?

Secondly, some parts of the entire article are somewhat lengthy and may need some structural adjustments.   In this study, a detailed analysis of the GATA gene family transcription factors was conducted. However, some of them could be placed in the supplementary materials, such as the Figure1-3. The content and relative expression of SOD, POD, and CAT were detected in the Figure 6 and Figure 9. To facilitate a better understanding of these information, these two figures could be changed into two consecutive figures. Gene expression analysis comes first, followed by enzyme activity analysis.

Here are some minor comments to this paper.

1.        There were three papers about the genome-wide identification of GATA transcription factors in potato. But the discussion of this papers is missing.

2.        Line 34 “located” should be “is located”.

3.        Line 117 “lengths” should be “length”.

4.        Line 224-237: Is there a mistake here? Shouldn't it refer to the transgenic plants of StGATA12 instead of StMAPKK5?

5.        In the Figure 5B-F, the legend for the different colored bar charts is missing.

6.        In Figure 8, Chlorophyll content was detected. Which specific chlorophyll content needs to be added to the figure legend?

Author Response

Comments and Suggestions for Authors

The production of potato is threatened by salinity. Potato plant growth is inhibited under salt stress due to ion toxicity caused by osmotic stress. This study identified the GATAs in potato and investigates the role of StGATA12, in response to salinity and osmotic stress. The findings indicate that StGATA12 enhances the growth and photosynthetic performance of potato plants under stress conditions by regulating key physiological and biochemical responses, suggesting its potential for improving the plant's resilience to adverse environmental conditions. These findings are significant for improving potato yield and quality. Here are some comments about this article.

Firstly, some experiments need to be improvement. In Figure 5, the data presented maybe from the in vitro seedlings. How about the phenotypes of these transgenic plants in the growth chamber?

Answer: Thanks for your question. Regarding Figure 5, you are correct that the data presented was obtained from in vitro seedlings. We are currently evaluating the phenotypes of the transgenic plants in the growth chamber and will include those results in our future updates. This will provide a more comprehensive understanding of their performance under controlled conditions.

Secondly, some parts of the entire article are somewhat lengthy and may need some structural adjustments.   In this study, a detailed analysis of the GATA gene family transcription factors was conducted. However, some of them could be placed in the supplementary materials, such as the Figure1-3. The content and relative expression of SOD, POD, and CAT were detected in the Figure 6 and Figure 9. To facilitate a better understanding of these information, these two figures could be changed into two consecutive figures. Gene expression analysis comes first, followed by enzyme activity analysis.

Answer: Figures 1-3 have been included as supplementary materials. Regarding Figure 9, the investigation of StSOD, StCAT, StPOD, and StP5CS is considered a downstream study. This exploration serves as a foundational finding for our future research endeavors.

Here are some minor comments to this paper.

  1. There were three papers about the genome-wide identification of GATA transcription factors in potato. But the discussion of this papers is missing.

Answer: In relation to the genome-wide identification of GATA TF in potato, we have included a discussion that compares our results with those from the three papers you mentioned.

  1. Line 34 “located” should be “is located”.
  2. Line 117 “lengths” should be “length”.
  3. Line 224-237: Is there a mistake here? Shouldn't it refer to the transgenic plants of StGATA12 instead of StMAPKK5?

Answer to q2, q3, and q4, the incorrect wording has been revised.

  1. In the Figure 5B-F, the legend for the different colored bar charts is missing.

Answer: we have added the legend.

  1. In Figure 8, Chlorophyll content was detected. Which specific chlorophyll content needs to be added to the figure legend?

Answer: Total chlorophyll content was assessed in this study and is expressed in milligrams per gram of fresh weight.

Reviewer 2 Report

Comments and Suggestions for Authors

Comments and Suggestions for Authors

The authors address an important problem from the perspective of global agriculture. They analyze two common abiotic stress factors and their effects on potato plants. The work I reviewed required much lab work and numerous bioinformatic analyses.

I probably did not understand this part 4.8 Please give a more detailed description of which leaves were collected by 4.11. - In physiological evaluation, e.g., chlorophyll measurement, in my experience, there are statistically significant differences in the chlorophyll content of different leaves collected from the same plant, depending on the plant's location and light intensity. Or were these perhaps leaves from seedlings whose photos can be seen in Fig. 5? How many of these seedlings were collected for analysis in this case? How much did a single leaf sample used for the biochemical analyses weigh?

In general, the methods are poorly described, e.g. H2O2 determination: I could not find the cited publication [64], but e.g. in this one: Alexieva, V., Sergiev, I., Mapelli, S. and Karanov, E. (2001), The effect of drought and ultraviolet radiation on growth and stress markers in pea and wheat. Plant, Cell & Environment, 24: 1337-1344. https://doi.org/10.1046/j.1365-3040.2001.00778.x, the method is adequately described; I ask the authors for more care and appropriate changes to this part of the manuscript.

Fig. 6 The scale in the corresponding graphs could be the same, e.g., CAT activity: NaCl up to 500u and PEG up to 400u, why can not it be up to 500u in both graphs? It is similar with the H2O2 content and Fig 8 ion leakage,

Please complete:

Author Contributions:

Funding:

Institutional Review Board Statement:

Informed Consent Statement:

Data Availability Statement:

Conflicts of Interest:

Line 67 “oxygen species” should be “reactive oxygen species”

Line 91 (line 388) “chlorophyll biogenesis” should be “chlorophyll biosynthesis”

Line 323 is a key regulatory enzyme involved in the biosynthesis of proline. – please quote this

Line 360 Numerous studies have reported that GATAs regulate plant growth and development previously. – Please quote this

Line 556: Typo: “he” – should probably be “The”

Fig. 1 and Fig. 2 are completely illegible; perhaps move them to the supplementary materials.

Fig. 5: There is no legend, it is not known what the individual bars mean

It is probably better to submit publications to a competent English teacher for review before publication. This will help avoid certain linguistic awkwardness and facilitate the inclusion of valuable content. Example:

Line: 364-365 “In this study, we found that StGATA12 gene also functions in the control of plant growth under normal conditions.”

That could be: In this study, we found that the StGATA12 gene also controls plant growth under normal conditions.

Line 398 "..., and increased antioxidant enzymes.” – What do the authors mean by this?

The discussion is very short and does not end with a summary. Please end the discussion with a summary.

The title 4.2 is misleading, and it would probably be better to write: bioinformatic predictions of biochemical properties and subcellular localization or even better to combine 4.1 – 4.7 into one chapter and call it e.g. “bioinformatic analyses”

Comments on the Quality of English Language

-

Author Response

Comments and Suggestions for Authors

The authors address an important problem from the perspective of global agriculture. They analyze two common abiotic stress factors and their effects on potato plants. The work I reviewed required much lab work and numerous bioinformatic analyses.

Answer: Further analysis of gene function and bioinformatic data will be performed in our future study.

I probably did not understand this part 4.8 Please give a more detailed description of which leaves were collected by 4.11. - In physiological evaluation, e.g., chlorophyll measurement, in my experience, there are statistically significant differences in the chlorophyll content of different leaves collected from the same plant, depending on the plant's location and light intensity. Or were these perhaps leaves from seedlings whose photos can be seen in Fig. 5? How many of these seedlings were collected for analysis in this case? How much did a single leaf sample used for the biochemical analyses weigh?

Answer: All leaves collected were the fully expanded third leaf from the top of the plant, gath-ered between 9:30 and 11:30 AM, including evaluation of physiological parameters, photosynthetic gas exchange parameters, enzyme activities, chlorophyll contents, and gene expression. In the analysis of StGATA expression in different tissues, potato leaves were collected from plants at the flowering stage.

For assessing StGATA expression in response to salinity and osmotic stress, leaves were collected at 0, 1, 3, 6, 12, and 24 hours after 4-week-old seedlings were exposed to NaCl and PEG treatment. To evaluate physiological parameters, including photosynthetic gas exchange, enzyme activities, chlorophyll content, and gene expression, leaves were collected 24 hours after the same seedlings were subjected to NaCl and PEG treatment. Additionally, plant phenotypes and growth were analyzed 4 weeks after the seedlings, aged 4 weeks and 6 days, were exposed to NaCl and PEG treatment. For the analysis of chlorophyll content, leaf samples were collected from a total of 5 seedlings. In the biochemical analyses, each leaf weighed approximately 0.02 g.

In general, the methods are poorly described, e.g. H2O2 determination: I could not find the cited publication [64], but e.g. in this one: Alexieva, V., Sergiev, I., Mapelli, S. and Karanov, E. (2001), The effect of drought and ultraviolet radiation on growth and stress markers in pea and wheat. Plant, Cell & Environment, 24: 1337-1344. https://doi.org/10.1046/j.1365-3040.2001.00778.x, the method is adequately described; I ask the authors for more care and appropriate changes to this part of the manuscript.

Fig. 6 The scale in the corresponding graphs could be the same, e.g., CAT activity: NaCl up to 500u and PEG up to 400u, why can not it be up to 500u in both graphs? It is similar with the H2O2 content and Fig 8 ion leakage,

Answer: The improper citation has been corrected. We appreciate your suggestion to standardize the scales in the corresponding graphs. The discrepancy in the maximum values for CAT activity (NaCl at 500 µM and PEG at 400 µM) was intended to reflect the different experimental conditions and results observed for these treatments.

Please complete:

Author Contributions:

Funding:

Institutional Review Board Statement:

Informed Consent Statement:

Data Availability Statement:

Conflicts of Interest:

Answer: The ‘Institutional Review Board Statement’ and ‘Informed Consent Statement’ are not applicable. The additional information has been included.

Line 67 “oxygen species” should be “reactive oxygen species”

Line 91 (line 388) “chlorophyll biogenesis” should be “chlorophyll biosynthesis”

Line 323 is a key regulatory enzyme involved in the biosynthesis of proline. – please quote this

Line 360 Numerous studies have reported that GATAs regulate plant growth and development previously. – Please quote this

Line 556: Typo: “he” – should probably be “The”

Answer: We have revised the aforementioned questions.

Fig. 1 and Fig. 2 are completely illegible; perhaps move them to the supplementary materials.

Answer: We have revised figure 1-3 and included then as the supplementary materials.

Fig. 5: There is no legend, it is not known what the individual bars mean

Answer: the legend has been provided.

It is probably better to submit publications to a competent English teacher for review before publication. This will help avoid certain linguistic awkwardness and facilitate the inclusion of valuable content. Example:

Line: 364-365 “In this study, we found that StGATA12 gene also functions in the control of plant growth under normal conditions.”

That could be: In this study, we found that the StGATA12 gene also controls plant growth under normal conditions.

Answer: we have engaged an English teacher to refine the language, resulting in significant improvements in clarity and quality.

Line 398 "..., and increased antioxidant enzymes.” – What do the authors mean by this?

Answer: we have revised it as follows: The transgenic lines with significantly increased StGATA12 gene expression exhibited reduced levels of oxidative products, along with increased antioxidative components and higher contents of antioxidant enzymes, compared to the non-transgenic plants.

The discussion is very short and does not end with a summary. Please end the discussion with a summary.

Answer: We have added the following summary: Conclusively, the StGATA12 gene is highly expressed in potato leaves subjected to salinity and osmotic stress in a concentration-dependent manner. Plants overexpressing GATA12 demonstrate normal growth, maintained photosynthetic capacity, and enhanced antioxidative ability when exposed to NaCl and PEG6000. Based on these findings, StGATA12 can be identified as a candidate gene for salt tolerance, which could facilitate the development and utilization of saline land affected by seawater intrusion. However, future studies should focus more on assessing tuber yield and quality. 

The title 4.2 is misleading, and it would probably be better to write: bioinformatic predictions of biochemical properties and subcellular localization or even better to combine 4.1 – 4.7 into one chapter and call it e.g. “bioinformatic analyses”

Answer: we have revised the title of section 4.2 to “bioinformatic predictions of biochemical properties and subcellular localization”.

Reviewer 3 Report

Comments and Suggestions for Authors

Dear Authors,

I have an opportunity to review manuscript entitled:” Genome-wide identification of GATA family genes in potato and characterization of StGATA12 in response to salinity and osmotic stress” submitted to IJMS MDPI.

Authors concentrated on potential role of GATA in potato plant responding to salt and osmotic stresses induced by NaCl and PEG. Authors revealed that expression of StGATA family genes were altered in response to salinity and osmotic stress. Moreover, StGATA12 overexpression induced biochemical responses of potato plants to salinity and osmotic stress by regulating the levels of H2O2, MDA, and proline as well as the activity of CAT, SOD, and POD. Furthermore, StGATA12 overexpression induced the up-regulation of StCAT, StSOD, StPOD, and StP5CS insalinity and osmotic stress condition.

Authors presenting logical and interesting studies, despite of it, some aspects need to be explained or corrected:

-introduction is well written and gives the reader efficient information to analyze Author’s obtained results, but the precise aim of studies should be added;

- figures 1 and 2 should be definitely enlarge, the important details are completely lost even in 200% enlargement;

-Did any reason to choose exactly cultivar Atlantic to these studies?

-factual error should be corrected: Authors was written “Tissue-specific expression of StGATA genes”, but taking information from M&M and results there are no tissues, for sure [leaf, petiole, stem…];

- Why Authors use only one reference gene?

- Please, use supplementary figures instead of “additional”; When Authors described in subchapter 2.6 GFP localization it should be added to the main manuscript; Besides of it- Please be aware in interpretation, because GFP control also gives signal in plasma membrane as well as in nucleus in Author’s microscopic photographs; Therefore, the significance of “increasing” of this localization is highly debatable; Moreover, Did this localization do in other plant tissues, not only in epidermis?

- figure 5 the same situation- phenotypic comparison is really hard in current version;

-Suddenly in the results appear StMAPKK5-overexpressing OE lines- Why we did not find any information about it in M&M and in discussion? Please, correct it; Is that phenotype of StGATA12 or StMAPKK5 OE ??

- figure 6 -please, enlarge it;

- did Authors test overexpression of StGATA57 and 46?

-Discussion part compared with obtained results is very short and firm- Please, add some future prospects coming from obtained results;

Author Response

Comments and Suggestions for Authors

Dear Authors,

I have an opportunity to review manuscript entitled:” Genome-wide identification of GATA family genes in potato and characterization of StGATA12 in response to salinity and osmotic stress” submitted to IJMS MDPI.

Authors concentrated on potential role of GATA in potato plant responding to salt and osmotic stresses induced by NaCl and PEG. Authors revealed that expression of StGATA family genes were altered in response to salinity and osmotic stress. Moreover, StGATA12 overexpression induced biochemical responses of potato plants to salinity and osmotic stress by regulating the levels of H2O2, MDA, and proline as well as the activity of CAT, SOD, and POD. Furthermore, StGATA12 overexpression induced the up-regulation of StCAT, StSOD, StPOD, and StP5CS insalinity and osmotic stress condition.

Authors presenting logical and interesting studies, despite of it, some aspects need to be explained or corrected:

-introduction is well written and gives the reader efficient information to analyze Author’s obtained results, but the precise aim of studies should be added;

Answer: We have added a more precise statement of the study’s aims at the end of the Introduction, which may enhance the reader’s understanding of the research context.

- figures 1 and 2 should be definitely enlarge, the important details are completely lost even in 200% enlargement;

Answer: These figures have been enlarged and improved for better clarity in the revised version.

-Did any reason to choose exactly cultivar Atlantic to these studies?

Answer: We selected the Atlantic cultivar for our studies because of its well-documented characteristics, including high yield and storability. This cultivar is widely grown in Guangzhou, China, where seawater intrusion has led to significant salinization in some areas of Guangdong province. Additionally, its common use in research facilitates better comparisons with existing literature. This choice enhances the relevance of our findings to broader agricultural practices and contributes to a deeper understanding of stress responses specific to this cultivar.

-factual error should be corrected: Authors was written “Tissue- specific expression of StGATA genes”, but taking information from M&M and results there are no tissues, for sure [leaf, petiole, stem…];

Answer: We recognized that the title “tissue-specific expression of StGATA genes” should be corrected to reflect that the study focuses on the expression in various organs, such as leaves, petioles, and stems, rather than specific tissues.

- Why Authors use only one reference gene?

Answer: In this study, StEf1a serves as an internal control for qRT-PCR. This is because of its stable expression level in different tissues and cells of potato plants, which has been confirmed in our preliminary experiment.

- Please, use supplementary figures instead of “additional”; When Authors described in subchapter 2.6 GFP localization it should be added to the main manuscript; Besides of it- Please be aware in interpretation, because GFP control also gives signal in plasma membrane as well as in nucleus in Author’s microscopic photographs; Therefore, the significance of “increasing” of this localization is highly debatable; Moreover, Did this localization do in other plant tissues, not only in epidermis?

Answer: we have revised our terminology to use “supplementary figures” instead of “additional”. It should be noticed that the expression level of endogenous GFP in plant cells is generally low. This means that, in most cases, the fluorescent signals observed in experiments using GFP are primarily due to the expression of transgenic constructs rather than any natural expression of GFP within the plant. As for your question about localization in other plant tissues, we will include the relevant findings in our future study.

- figure 5 the same situation- phenotypic comparison is really hard in current version;

Answer: Exactly, regardless of whether the plants are transgenic or non-transgenic, seedlings exposed to 150 mM NaCl and 20% PEG6000 exhibit growth inhibition. As a result, making comparisons is challenging.

-Suddenly in the results appear StMAPKK5-overexpressing OE lines- Why we did not find any information about it in M&M and in discussion? Please, correct it; Is that phenotype of StGATA12 or StMAPKK5 OE ??

Answer: We apologize for the error regarding StMAPKK5. It has been corrected to StGATA12.

- figure 6 -please, enlarge it;

Answer: We have enlarged it to enhance visibility and clarity.

- did Authors test overexpression of StGATA57 and 46?

Answer: Indeed, there is a significant increase in the expression of StGATA46 and StGATA57 in potato plants in response to NaCl and PEG6000 treatment. Establishing overexpressing lines for StGATA46 and StGATA57 is currently on our agenda.

-Discussion part compared with obtained results is very short and firm- Please, add some future prospects coming from obtained results;

Answer: We have integrated more detailed perspectives on potential applications and future research directions based on our findings at the end of the Discussion.

Round 2

Reviewer 1 Report

Comments and Suggestions for Authors

I have reviewed your revised manuscript, and I would like to commend you on the thoughtful revisions and thorough responses to the reviewers' comments. Your efforts have significantly strengthened the manuscript, and I am pleased with the improvements made.

 Maybe there is a  extra figure in page 5. I have no further substantive comments, as your responses have addressed the key concerns raised during the initial review.  Congratulations on your excellent work, and I look forward to seeing the final version published.

Reviewer 3 Report

Comments and Suggestions for Authors

The most important English correction were done and  improved understanding of the manuscript;  some figures imperfections were corrected, as well.

Factual errors were corrected. 

The problems regarding reference genes and fluorescence signal localisation and detection are still exist.